# Locally Most Powerful Bayesian Test for Out-of-Distribution Detection Using Deep Generative Models

**Keunseo Kim**   **Juncheol Shin**   **Heeyoung Kim**
Department of Industrial and Systems Engineering, KAIST
{kkseo91, jczy0402, heeyoungkim}@kaist.ac.kr

## Abstract

Several out-of-distribution (OOD) detection scores have been recently proposed for deep generative models because the direct use of the likelihood threshold for OOD detection has been shown to be problematic. In this paper, we propose a new OOD score based on a Bayesian hypothesis test called the locally most powerful Bayesian test (LMPBT). The LMPBT is locally most powerful in that the alternative hypothesis (the representative parameter for the OOD sample) is specified to maximize the probability that the Bayes factor exceeds the evidence threshold in favor of the alternative hypothesis provided that the parameter specified under the alternative hypothesis is in the neighborhood of the parameter specified under the null hypothesis. That is, under this neighborhood parameter condition, the test with the proposed alternative hypothesis maximizes the probability of correct detection of OOD samples. We also propose numerical strategies for more efficient and reliable computation of the LMPBT for practical application to deep generative models. Evaluations conducted of the OOD detection performance of the LMPBT on various benchmark datasets demonstrate its superior performance over existing OOD detection methods.

## 1   Introduction

In several real-world applications of deep learning models, detecting anomalous samples that significantly deviate from the distribution of the training data, that is, out-of-distribution (OOD) samples, is crucial for reliable decision making, and various OOD detection methods have been studied in this regard. Deep generative models [9, 8, 17] have also been investigated for OOD detection as an intuitive strategy owing to their ability to evaluate the likelihood of a test sample. However, recent studies have shown that deep generative models can assign higher likelihoods to OOD samples than in-distribution samples [12], causing the direct use of raw likelihoods for OOD detection to be problematic.

To address this challenge, several OOD scores have been recently proposed for deep generative models and have exhibited effective OOD detection performance [15, 16, 18]. Ren et al. [15] proposed the use of the ratio of the likelihood obtained from a model trained using pure input data to that obtained from a background model trained using noise-perturbed input data as an OOD score. Xiao et al. [18] proposed a likelihood regret score that can be calculated as the difference between the likelihood obtained with the optimized parameters for a test sample and that approximated by the VAE. Serrà et al. [16] proposed a penalized log-likelihood with an input complexity as an OOD score, where the input complexity can be computed as the normalized size of the compressed input image.

In this paper, we propose a new OOD detection method that is optimal in some sense in the framework of a Bayesian hypothesis test—inspired by a uniformly most powerful Bayesian test [7] and a locally

35th Conference on Neural Information Processing Systems (NeurIPS 2021).

most powerful test [14]. The Bayesian hypothesis test is based on the ratio of the posterior odds that the alternative hypothesis is true, given the observed data [3]. We formulate the OOD detection problem as a Bayesian test with the null hypothesis that a test sample is an in-distribution sample and the alternative hypothesis that the test sample is an OOD sample. Specifically, the null hypothesis is specified to represent the model trained on the in-distribution training set by the maximum likelihood estimation, as typically performed in previous studies. Against this specific null hypothesis, we propose to specify the alternative hypothesis to represent the model trained on the expanded training set with the test sample. Then, we show that the test with our proposed alternative hypothesis is locally most powerful in that the alternative hypothesis is specified so as to maximize the probability that the Bayes factor exceeds the evidence threshold in favor of the alternative hypothesis among all alternative hypotheses that have the model parameters in the neighborhood of the model parameter specified under the null hypothesis. That is, under this neighborhood parameter condition, the test with the proposed alternative hypothesis maximizes the probability of the correct detection of OOD samples. The proposed test is called the locally most powerful Bayesian test (LMPBT).

Because we specify the alternative hypothesis of the LMPBT for a test sample to represent the model trained on the expanded training set with that test sample, we need to re-train the model for each test sample. However, the computational cost of this retraining is significant. To address this issue, we adopt the upweighting method [10]. Using this method, we analyze the influence of upweighting a test sample on the parameter change and estimate the new parameters without retraining.

However, computational issues are encountered when the upweighting method is directly applied in practice. First, the upweighting method involves the Hessian matrix, which is assumed to have all positive eigenvalues. However, the loss function of deep generative models is known to be nonconvex, and finding the global optimum is infeasible. Consequently, the Hessian can have negative eigenvalues. Second, the upweighting method requires the computation of the inverse of the Hessian matrix. However, deep generative models, which have numerous parameters, incur large costs for the calculation of the Hessian and its inverse.

To address these issues, we use a low-rank approximation [2, 20, 21, 13] of the Hessian. Consequently, we can ensure that the approximated Hessian has all positive eigenvalues and can significantly reduce the computational cost for calculating the inverse of the Hessian. We evaluated the performance of the LMPBT using deep generative models on benchmark datasets and demonstrated a more effective OOD detection performance than competing methods. In summary, the contributions of this study are as follows:

- We propose a new OOD detection method—the locally most powerful Bayesian test (LMPBT)—that maximizes the probability of correct detection of OOD samples under some conditions.
- We address the computational issues encountered when practically implementing the LMPBT for deep generative models.
- Evaluations conducted of the LMPBT using variational autoencoders (VAEs) [9] and Glows [8] on benchmark datasets demonstrate state-of-the-art performance for OOD detection (the code for the LMPBT is available at https://github.com/keunseokim91/LMPBT).

## 2 Background: Deep Generative Models

Generally, deep probabilistic generative models comprise two sub-models: a generative model and an inference model. By introducing a latent variable $\mathbf{z}$, the generative model defines the generative process of $\mathbf{x}$ from $\mathbf{z}$, whereas the inference model infers the distribution of $\mathbf{x}$ given $\mathbf{z}$. For example, in flow-based generative models, the generative and inference models are defined using an invertible function $f$, which is composed of a sequence of invertible functions, $f = f_1 \circ f_2 \circ ... \circ f_k$. The generative and inference processes between $\mathbf{x}$ and $\mathbf{z}$ can be expressed as

$$\mathbf{x} \xleftrightarrow{f_1} \mathbf{h}_1 \xleftrightarrow{f_2} \mathbf{h}_2 \cdots \xleftrightarrow{f_k} \mathbf{z},$$

where the latent variable $\mathbf{z}$ has a tractable density $p_\theta(\mathbf{z})$. Then, the log probability density function of the model given a sample $\mathbf{x}$ is given by

$$\log p_\theta(\mathbf{x}) = \log p_\theta(\mathbf{z}) + \log \left| \det \left( \frac{\partial \mathbf{z}}{\partial \mathbf{x}} \right) \right| = \log p_\theta(\mathbf{z}) + \sum_{i=1}^{k} \log \left| \det \left( \frac{\partial \mathbf{h}_i}{\partial \mathbf{h}_{i-1}} \right) \right|, \qquad (1)$$

where we define $\mathbf{h}_0 = \mathbf{x}$ and $\mathbf{h}_k = \mathbf{z}$.

In VAEs, the generative and inference processes are modeled using the encoder–decoder structure. We assume that $\mathbf{x}|\mathbf{z} \sim p_\theta(\mathbf{x}|\mathbf{z})$ and $\mathbf{z} \sim p(\mathbf{z})$, where $p_\theta(\mathbf{x}|\mathbf{z})$ is parameterized using deep neural networks with parameters $\theta$ and $p(\mathbf{z})$ is a tractable prior distribution of $\mathbf{z}$. The maximum likelihood estimate (MLE) of $\theta$ requires maximizing the marginal likelihood $p_\theta(\mathbf{x}) = \int p_\theta(\mathbf{x}|\mathbf{z})p(\mathbf{z})d\mathbf{z}$. However, this computation is intractable, and variational inference is used to derive the evidence lower bound (ELBO) as follows:

$$\log p_\theta(\mathbf{x}) \geq E_{q_\phi(\mathbf{z}|\mathbf{x})}\left[\log p_\theta(\mathbf{x}|\mathbf{z}) - D_{\mathrm{KL}}(q_\phi(\mathbf{z}|\mathbf{x})||p(\mathbf{z}))\right] = \mathrm{ELBO}, \tag{2}$$

where the variational posterior $g_\phi(\mathbf{z}|\mathbf{x})$ approximates the true posterior $p_\theta(\mathbf{z}|\mathbf{x})$ and is parameterized by deep neural networks with variational parameters $\phi$ and $D_{\mathrm{KL}}$ is the Kullback–Liebler (KL) divergence. By Eq. (2), the ELBO may be used as an approximation of the log-likelihood. Using the exact likelihood in Eq.(1) and the approximated likelihood in Eq.(2), respectively, flow-based generative models and VAEs are typically trained by minimizing the negative log-likelihood of the training dataset $\{\mathbf{x}_1, \mathbf{x}_2, \ldots, \mathbf{x}_n\}$ as a training loss:

$$-\sum_{i=1}^{n} \log L(\theta|\mathbf{x}_i) = -\sum_{i=1}^{n} \log p_\theta(\mathbf{x}_i). \tag{3}$$

## 3    Locally Most Powerful Bayesian Test for OOD Detection

We formulate the OOD detection problem as a Bayesian hypothesis test. We compare the null hypothesis $H_0$ that a test sample $\mathbf{x}_t$ is an in-distribution sample and the alternative hypothesis $H_1$ that the test sample $\mathbf{x}_t$ is an OOD sample. Specifically, we consider simple null and alternative hypotheses as follows:

$$
\begin{aligned}
H_0 &: \theta = \theta_0, \\
H_1 &: \theta = \theta_1,
\end{aligned}
\tag{4}
$$

where $\theta_0$ and $\theta_1$ are representative parameters of a deep generative model for the in-distribution and OOD samples, respectively, and $\theta$ is a set of parameters from the parameter space $\Theta$.

The Bayesian hypothesis test is based on the ratio of the posterior odds that the alternative hypothesis is true given the observed data [3]. The posterior odds in favor of the alternative hypothesis equals the Bayes factor multiplied by the prior odds in favor of the alternative hypothesis. When the null and alternative hypotheses are both simple, the Bayes factor represents the ratio of the likelihoods of the data evaluated under the two hypotheses. The posterior odds for our hypothesis test in Eq.(4) for the test sample $\mathbf{x}_t$ can be computed as

$$\frac{P(H_1|\mathbf{x}_t)}{P(H_0|\mathbf{x}_t)} = \frac{L(\theta_1|\mathbf{x}_t)}{L(\theta_0|\mathbf{x}_t)} \times \frac{P(H_1)}{P(H_0)}. \tag{5}$$

Here, $P(H_i)$ is the prior probability for hypothesis $H_i$, and $L(\theta_1|\mathbf{x}_t)/L(\theta_0|\mathbf{x}_t)$ is the Bayes factor in favor of the alternative hypothesis, where $L(\theta_1|\mathbf{x}_t)$ and $L(\theta_0|\mathbf{x}_t)$ are the likelihoods of $\mathbf{x}_t$ under $H_1$ and $H_0$, respectively.

To complete the test, we must specify $\theta_0$ and $\theta_1$, and the prior probabilities for $H_0$ and $H_1$. Typically, $\theta_0$ is specified as the MLE, denoted by $\hat{\theta}$, which maximizes the log-likelihood function of the training dataset $\{\mathbf{x}_1, \mathbf{x}_2, \ldots, \mathbf{x}_n\}$ as follows:

$$\hat{\theta} = \operatorname*{argmax}_{\theta \in \Theta} \sum_{i=1}^{n} \log L(\theta|\mathbf{x}_i). \tag{6}$$

However, the specification of $\theta_1$ is difficult owing to the lack of information on the OOD samples. In fact, the previous likelihood-ratio-based OOD detection methods [15, 16, 18] can be expressed using the hypothesis test in Eq.(4) with different specifications of $\theta_1$. The OOD detection performance is affected by the specification of $\theta_1$; thus, it is important to choose $\theta_1$ optimally.

### 3.1 Proposed OOD score

In this paper, we propose a theoretically grounded method to specify $\theta_1$ optimally in some sense. Specifically, we specify $\theta_1$ as the MLE of the expanded training set with the test sample $\mathbf{x}_t$, denoted as $\hat{\theta}_t$, as follows:

$$\hat{\theta}_t = \underset{\theta}{\operatorname{argmax}} \left[ \log L(\theta|\mathbf{x}_t) + \sum_{i=1}^{n} \log L(\theta|\mathbf{x}_i) \right]. \tag{7}$$

Then, for each test sample $\mathbf{x}_t$, we need to solve the optimization problem in Eq.(7) to obtain $\hat{\theta}_t$. However, retraining the model for each test sample is computation-intensive. Instead, we use a computational approximation for the solution of Eq.(7), which allows us to obtain $\hat{\theta}_t$ easily using the already obtained $\hat{\theta}$. Specifically, we adopt the upweighting method [10] for the approximation. This method analyzes the influence of a certain training data point on the parameter change by upweighting the loss of that point. It is known that the influence of upweighting the loss of a training data point $\mathbf{x}$ by scale $\epsilon$ is given by

$$\frac{d\hat{\theta}_t}{d\epsilon}|_{\epsilon=0} = -H_{\hat{\theta}}^{-1} \nabla_\theta(-\log L(\hat{\theta}|\mathbf{x})), \tag{8}$$

where $H_{\hat{\theta}} = \frac{1}{n} \sum_{i=1}^{n} \nabla_\theta^2(-\log L(\hat{\theta}|\mathbf{x}_i))$ is the Hessian at $\hat{\theta}$ and $\nabla_\theta(-\log L(\hat{\theta}|\mathbf{x}))$ is the loss gradient at $\mathbf{x}$ with respect to $\theta$. In our case, the training dataset $X_n$ does not contain the test sample $\mathbf{x}_t$; thus, the weight of $\mathbf{x}_t$ in the training loss in Eq.(3) can be regarded as zero. Then, adding the loss of $\mathbf{x}_t$ to the training loss for $X_n$ is equivalent to upweighting the loss of $\mathbf{x}_t$ by $1/n$. Finally, $\hat{\theta}_t$ can be approximated as

$$\hat{\theta}_t \approx \tilde{\theta}_t = \hat{\theta} - \frac{1}{n} H_{\hat{\theta}}^{-1} \nabla_\theta(-\log L(\hat{\theta}|\mathbf{x}_t)). \tag{9}$$

Finally, with the specification of $\theta_0 = \hat{\theta}$ and $\theta_1 = \tilde{\theta}_t$ with the additional assumption of equal prior probabilities for $H_0$ and $H_1$ (i.e., $P(H_1) = P(H_2) = 0.5$), we propose the log of the posterior odds in Eq.(5) as an OOD score for test sample $\mathbf{x}_t$, denoted by $S(\mathbf{x}_t)$, as follows:

$$S(\mathbf{x}_t) = -\log L(\hat{\theta}|\mathbf{x}_t) + \log L(\tilde{\theta}_t|\mathbf{x}_t). \tag{10}$$

In the next section, we show that the test using the proposed OOD score in Eq.(10) maximizes the probability that the alternative hypothesis is accepted when it is true (i.e., the probability of the correct OOD sample detection) provided that the parameter specified under the alternative hypothesis is in the neighborhood of the parameter specified under the null hypothesis.

### 3.2 Locally most powerful Bayesian test for OOD detection

Inspired by the uniformly most powerful Bayesian test [7] and the locally most powerful test [14], we define the locally most powerful Bayesian test (LMPBT) for OOD detection.

**Definition 3.1** (Locally most powerful Bayesian test (LMPBT) for OOD detection). Consider the OOD detection test in Eq. (4) for test sample $\mathbf{x}_t$. Let $\theta_*$ be the true parameter associated with $\mathbf{x}_t$. The locally most powerful Bayesian test for evidence threshold $\gamma > 0$, denoted by LMPBT($\gamma$), in favor of the alternative hypothesis $H_1 : \theta = \theta_1$ against a fixed null hypothesis $H_0 : \theta = \theta_0$, is a Bayesian hypothesis test in which the Bayes factor for the test satisfies the following inequality for any $\theta_* \in \Theta$ and all alternative hypotheses $H_1' : \theta = \theta_1'$, under the condition $0 < d(\theta_1', \theta_0) < \delta$, where $\delta$ is a sufficiently small constant and $d$ is a measure of the distance:

$$P_{\theta_*} \left[ \frac{L(\theta_1|\mathbf{x}_t)}{L(\theta_0|\mathbf{x}_t)} > \gamma \right] \geq P_{\theta_*} \left[ \frac{L(\theta_1'|\mathbf{x}_t)}{L(\theta_0|\mathbf{x}_t)} > \gamma \right]. \tag{11}$$

That is, with a fixed null hypothesis $H_0 : \theta = \theta_0$, LMPBT($\gamma$) is a Bayesian test in which the alternative hypothesis $H_1 : \theta = \theta_1$ is specified to maximize the probability that the Bayes factor $L(\theta_1|\mathbf{x}_t)/L(\theta_0|\mathbf{x}_t)$ exceeds the evidence threshold $\gamma$ for all possible values of the true parameter $\theta_*$ associated with test sample $\mathbf{x}_t$, under the condition that $\theta_1$ is in the neighborhood of $\theta_0$.

**Proposition 1.** *The test using the OOD score in Eq.(10) is LMPBT($\gamma$), given a sufficiently large training dataset.*

*Proof.* We prove that, in our OOD detection test with the specification of $\theta_0 = \hat{\theta}$ and $\theta_1 = \tilde{\theta}_t$ for the test sample $\mathbf{x}_t$, the Bayes factor $L(\tilde{\theta}_t|\mathbf{x}_t)/L(\hat{\theta}|\mathbf{x}_t)$ satisfies the inequality in Eq.(11) by showing that the Bayes factor in favor of the alternative $H_1 : \theta = \theta_1$ (i.e., $L(\theta_1|\mathbf{x}_t)/L(\hat{\theta}|\mathbf{x}_t)$) is maximized when $\theta_1 = \tilde{\theta}_t$. Under the condition that $\theta_1$ is in the neighborhood of $\theta_0 = \hat{\theta}$, we can use the Taylor expansion of $\log L(\theta_1|\mathbf{x}_t)$ at $\hat{\theta}$ as follows:

$$\log L(\theta_1|\mathbf{x}_t) \approx \log L(\hat{\theta}|\mathbf{x}_t) + (\theta_1 - \hat{\theta})^T \nabla_\theta \log L(\hat{\theta}|\mathbf{x}_t) + \frac{1}{2}(\theta_1 - \hat{\theta})^T(-H_{\hat{\theta}})(\theta_1 - \hat{\theta}) + R(\theta_1).$$

Here, $R(\theta_1)$ is the remainder term, which has an upper bound $\frac{M}{6}||\theta_1 - \hat{\theta}||^3$ (Lagrange error bound [6]), where $M$ is an upper bound of the third derivative of the log-likelihood for $||\theta_1 - \hat{\theta}|| < \delta$. With a sufficiently small $\delta$, the first and second terms will dominate the remainder term because higher-order terms vanish much more rapidly, and the remainder term will be negligible. Then, the log-Bayes factor can be approximated as

$$\log L(\theta_1|\mathbf{x}_t) - \log L(\hat{\theta}|\mathbf{x}_t) \approx (\theta_1 - \hat{\theta})^T \nabla_\theta \log L(\hat{\theta}|\mathbf{x}_t) + \frac{1}{2}(\theta_1 - \hat{\theta})^T(-H_{\hat{\theta}})(\theta_1 - \hat{\theta}). \quad (12)$$

To obtain $\theta_1$ that maximizes the log-Bayes factor, we find $\theta_1$ that makes the gradient of Eq. (12) equal to zero. That is,

$$\nabla_\theta[\log L(\theta_1|\mathbf{x}_t) - \log L(\hat{\theta}|\mathbf{x}_t)] \approx \nabla_\theta \log L(\hat{\theta}|\mathbf{x}_t) + (\theta_1 - \hat{\theta})^T(-H_{\hat{\theta}}) = 0. \quad (13)$$

Subsequently, the solution to Eq. (13) is obtained as $(\theta_1 - \hat{\theta}) = -H_{\hat{\theta}}^{-1}\nabla_\theta(-\log L(\hat{\theta}|\mathbf{x}_t))$. To ensure the condition $0 < d(\theta_1, \hat{\theta}) < \delta$ for sufficiently small $\delta$, we rescale $(\theta_1 - \hat{\theta})$ with a sufficiently small constant $\epsilon$ as $(\theta_1 - \hat{\theta}) = -\epsilon H_{\hat{\theta}}^{-1}\nabla_\theta(-\log L(\hat{\theta}|\mathbf{x}_t))$. By setting $\epsilon = \frac{1}{n}$ with a sufficiently large $n$, we can obtain the desired result:

$$\theta_1 = \hat{\theta} - \frac{1}{n}H_{\hat{\theta}}^{-1}\nabla_\theta(-\log L(\hat{\theta}|\mathbf{x}_t)) = \tilde{\theta}_t.$$

$\square$

In Definition 3.1, the Bayesian hypothesis test requires a predefined evidence threshold $\gamma$. Practically, we can set the threshold as the specified (e.g., fifth) percentile in the distribution of the LMPBT score evaluated on the training set. Specifically, we can obtain the empirical distribution of the LMPBT score using "in-distribution" training samples. Then, we set a specified percentile in the distribution of the LMPBT score as the threshold, depending on the desired false positive rate. For example, we can set the fifth percentile in the distribution of the LMPBT score as the threshold, based on the fact that a false positive rate of 0.05 is typically used as a default value in practice.

## 4 Computational Issues on the LMPBT

### 4.1 Low-rank approximation of the Hessian

When using the upweighting method in Eq.(9), $H_{\hat{\theta}}$ is assumed to be positive definite; that is, $H_{\hat{\theta}}$ is assumed to have all positive eigenvalues. However, the loss function of deep neural networks is known to be non-convex, and finding the global optimum is infeasible. In fact, gradient-descent (GD) methods, which are generally used for deep generative models, are only guaranteed to find first-order stationary points, including saddle points [19]. Theoretically, GD methods are always able to escape saddle points, but they can take an exponentially long time [4]. In practice, $\hat{\theta}$ obtained by running GD methods with early stopping differs from the global optimum and could be a saddle point. Consequently, $H_{\hat{\theta}}$ could have negative eigenvalues.

To address this issue, we use a low-rank approximation of the Hessian utilizing eigenvalue decomposition [13]. Recent empirical studies show that the Hessians of neural networks are typically of a low rank and often have at least one negative eigenvalue [1]. We empirically demonstrate this phenomenon in the deep generative models in Figure 1, which shows the distribution of the eigenvalues of $H_{\hat{\theta}}$ calculated from a Glow trained on the CIFAR-10 dataset (left) and a VAE trained on the CIFAR-10 dataset (right). We can observe that most of the eigenvalues are zero, which indicates

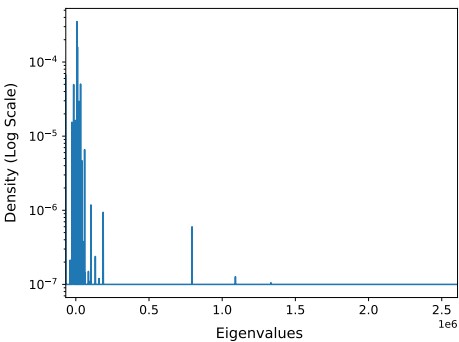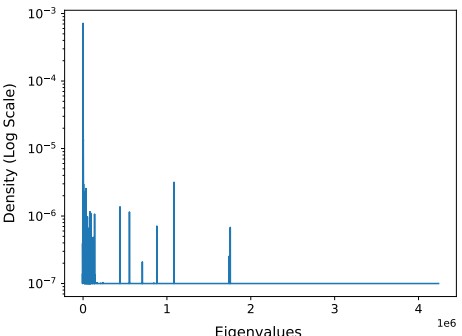

Figure 1: Distribution of the eigenvalues of $H_{\hat{\theta}}$ calculated from a Glow trained on the CIFAR-10 dataset (left) and a VAE trained on the CIFAR-10 dataset (right). We used the stochastic Lanczos quadrature algorithm [20] to compute the full empirical spectral density of the eigenvalues. It is notable that most of the eigenvalues are centered around zero and negative eigenvalues exist.

that the Hessian has a low rank. Further, the Hessian has negative eigenvalues. Benefiting from the low-rank property of the Hessian of deep generative models, we approximate the Hessian using a low-rank approximation with the top $r$ eigenvalues. Specifically, let the eigenvalue decomposition of the Hessian be given by

$$H = U\Gamma U^T = \sum_{i=1}^{p} \gamma_i u_i u_i^T, \tag{14}$$

where $p$ is the number of parameters, eigenvalues $\gamma_i$ are sorted with respect to their magnitudes—that is, $\gamma_i > \gamma_j$ for $i < j$—and $u_i$ are the corresponding eigenvectors. A low-rank approximation of the Hessian of rank $r$ is as follows:

$$\tilde{H} = U_r \Gamma_r U_r^T = \sum_{i=1}^{r} \gamma_i u_i u_i^T. \tag{15}$$

In practice, we must determine the appropriate value of the hyperparameter $r$ for each experiment. In our experiments in Section 5.2, we investigate the change in the performance of the LMPBT with different specifications of $r$. Even the models with millions of parameters exhibit effective OOD performance using a low-rank approximation of the Hessian with only the top 20∼30 eigenvalues.

## 4.2 Scalability

The computation of the Hessian and its inverse incurs costs proportional to the square and cube of the model parameter size, respectively. For deep generative models with numerous parameters, the computation of Eq. (9), which involves the inverse of the Hessian, is not feasible. We can overcome this issue by computing the Hessian-vector product, rather than explicitly forming the Hessian matrix. The Hessian-vector product can be computed as follows [20]:

$$\frac{\partial g_\theta^T z}{\partial \theta} = \frac{\partial g_\theta^T}{\partial \theta} z + g_\theta^T \frac{\partial z}{\partial \theta} = \frac{\partial g_\theta^T}{\partial \theta} z = \mathbf{H}z, \tag{16}$$

where $g_\theta = \nabla_\theta(-\log L(\theta|\cdot))$ is the gradient with respect to $\theta$, and $\mathbf{z}$ is a random vector. In Eq. (16), the first equality is obtained from the chain rule, the second equality from the independence of $\mathbf{z}$ and $\theta$, and the third equality from the definition of the Hessian. It is notable that the Hessian-vector product only incurs a cost that is the same as that of one gradient backpropagation. Then, using the Hessian-vector product results, the top $r$ eigenvalues and the corresponding eigenvectors of the Hessian and, hence, $\tilde{H}$ in Eq. (15), can be easily obtained using power iteration [20].

Another advantage of using the low-rank approximation of the Hessian in Eq. (15) is that we can easily compute Eq. (9), rather than directly computing the inverse of the Hessian, by utilizing the following:

$$\tilde{H}_{\hat{\theta}}^{-1} \nabla_\theta(-\log L(\hat{\theta}|\mathbf{x}_t)) = -\sum_{i=1}^{r} \frac{1}{\gamma_i} u_i u_i^T \nabla_\theta \log L(\hat{\theta}|\mathbf{x}_t). \tag{17}$$

Similar to the Hessian-vector product, Eq. (17) can be computed without computing the outer product $u_i u_i^T$, and it only incurs a computational cost proportional to the model parameter size.

## 5 Experimental Evaluation

In this section, we evaluate the performance of the LMPBT. Specifically, we show that the LMPBT yields superior OOD detection performance compared to competing OOD detection methods on various benchmark datasets.

### 5.1 Experimental setup

**Datasets** We used six benchmark image datasets—namely, MNIST, FASHION-MNIST (FMNIST), CelebA, CIFAR-10, CIFAR-100, and SVHN—and two synthetic datasets: Noise and Constant. To generate the Noise and Constant datasets, we adopted the sampling methods employed in [18]. We resized the images such that their width and height were 32 pixels. In the experiments, we used three datasets, namely, FMNIST, CIFAR-10, and CIFAR-100, as the training datasets. Specifically, we used the training partitions of the datasets for training. In the experiments using FMNIST, three-channel images were converted to grayscale to match the number of channels with one-channel FMNIST images. In the experiments using CIFAR-10 and CIFAR-100, one-channel images were duplicated three times to match the number of channels with three-channel CIFAR-10 and CIFAR-100 images. After training the deep generative models on the FMNIST, CIFAR-10, and CIFAR-100 datasets, we used the test partition of other datasets for testing to evaluate the OOD detection performance.

**Base deep generative models** We selected two representative deep generative models, VAE [9] and Glow [8], to evaluate the performance of the LMPBT. To train the VAE, we used the Adam optimizer to maximize the ELBO in Eq. (2). For the Glow, we adopted the structure used in [8] and used the Adam optimizer to maximize the log-likelihood in Eq. (1). To approximate the Hessian, we used the top 20 eigenvalues for the Glow and top 30 eigenvalues for the VAE. We discuss the specification of the number of eigenvalues in more detail in the following section. In the VAE experiments, we fixed the decoder and calculated the Hessian with respect to the encoder. The experimental settings and network structures are described in Supplementary Material.

**Metrics** To quantitatively evaluate the OOD detection performance, we employed the area under the receiver operating characteristic curve (AUROC), area under the precision-recall curve (AUPR) [5], and false positive rate at 80% true positive rate (FPR80) as evaluation metrics [15].

**Baseline OOD detection methods for comparison** We employed deep generative model-based OOD detection methods as baselines for comparison. These include the likelihood regret (LR) score [18], likelihood ratio (LLR) score [15], and input complexity (IC) method [16], as discussed in Section 1. These OOD detection methods were evaluated for both the Glow and VAE. For the VAE, we re-implemented the experiments according to the experimental settings in [18]. For the Glow, we also re-implemented the experiments according to the experimental settings in the original papers of the LLR [15] and IC [16]. Specifically, we used a PNG compressor to measure the input complexity when computing the IC. When computing the LR score for each test sample, we started with the trained encoder of a VAE or the trained Glow, and optimized its parameters for 100 steps.

### 5.2 Sensitivity and computational cost analysis for the LMPBT

Before the sensitivity analysis, we first visualize the OOD detection ability of the LMPBT, similar to [12]. Figure 2 (left) shows the histogram of the LMPBT scores when a VAE trained on CIFAR-10 was tested on CIFAR-10 as an in-distribution dataset and SVHN as an OOD dataset. We can observe that, if the LMPBT scores are higher, the corresponding test samples are more likely to be classified as OOD samples. As desired, the SVHN test samples (i.e., OOD samples) tended to be assigned higher LMPBT scores than the CIFAR-10 test samples.

**Specification of low-rank approximation** Figure 2 (b) shows the OOD detection performance (AUROC, AUPR, and FPR80) of the LMPBT according to the number of eigenvalues used for a low-rank approximation of the Hessian. The experiments were conducted using a VAE trained on CIFAR-10 and tested on CIFAR-10 and SVHN. Although the model has 1.4 million parameters, it achieved the best performance using only the top 30 eigenvalues. With less than 10 eigenvalues, the LMPBT

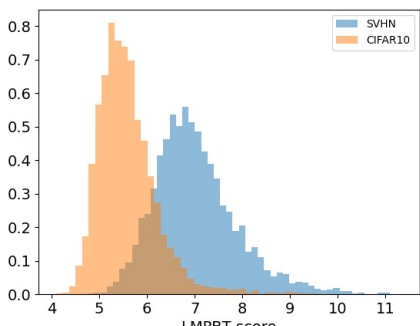 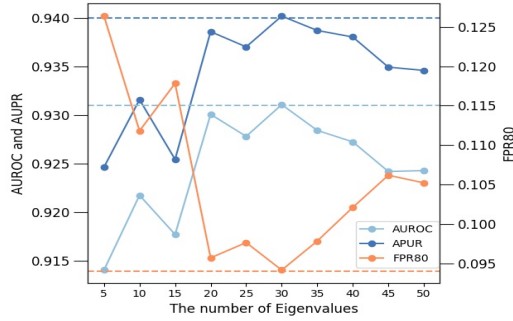

Figure 2: (left) LMPBT scores obtained with a VAE trained on CIFAR-10 and tested on both CIFAR-10 and SVHN. (right) OOD detection performance (AUROC, AUPR, and FPR80) versus the number of eigenvalues.

effectively detected OOD samples (AUROC: 0.913). This was possible because the Hessian was sparse. Interestingly, the performance does not increase monotonically with the number of eigenvalues. This behavior may be explained by the identifiability issue of the overparameterized models. For overparameterized models, there are often "degenerate" directions present in the parameter space [11], which correspond to the smallest eigenvalues of the Hessian. The change in the parameters in degenerate directions leads to almost no difference in the outputs and causes an identifiability issue. Deep generative models can also suffer from this identifiability issue. However, this issue can be obviated by using a low-rank approximation of the Hessian with the largest eigenvalues. Because changes in the output values in the degenerate directions can be considered random noise, the test accuracy of a compressed model can sometimes be better than that of the full model.

**Computational cost** The LMPBT requires the computation of the gradient and Hessian-vector product for each test sample, whereas the LLR and IC only require likelihood computation. Meanwhile, the LR requires several iterations for the optimization for each test sample. To determine whether the computational cost of the LMPBT is acceptable, we compared it with that of the LR. In our experiments using a GeForce RTX 2080 GPU and a VAE trained on CIFAR-10, on average, the LMPBT and LR required 0.11s and 0.30s to test a single sample, respectively. This demonstrates that the computational cost of the LMBPT is acceptable, although it is higher than those of the LLR and IC, which required 0.0125 s and 0.0128 s on average to test a single sample, respectively.

Moreover, to evaluate the computational cost for computing the top $r$ Hessian eigenvectors, we performed two sets of additional experiments. First, we compared the computational cost required for the Hessian eigenvalue decomposition with that for network parameter estimation, which can be considered a baseline because network parameter estimation is equally performed for other OOD detection methods. In our additional experiments using FMNIST ($1\times32\times32$) with VAEs, it took 2543 s to obtain the top 30 eigenvectors. This cost is relatively small (approximately 18%) compared with the cost of network parameter estimation (13448 s).

Second, we evaluated the scalability of Hessian eigenvalue decomposition. Specifically, we measured the computational cost for the eigenvalue decomposition of the Hessian (top 30 eigenvectors) for the datasets of various sizes with VAEs. We also calculated the ratio of the time cost to the number of parameters to investigate the relationship between the dimension of the Hessian and the computational cost. The results are listed in Table 1. The ratios obtained were 0.0028, 0.0023, and 0.0025 s for various data sizes. The ratio remained similar, which indicates that the computational cost tends to increase linearly with the number of parameters. These results show that our method is scalable and can be applied to large-scale networks.

### 5.3 Performance comparison with other OOD scores

To evaluate the LMPBT, we compared the ROC curve obtained using the LMPBT with those using the LR, IC, and LLR for OOD detection with a VAE trained on CIFAR-100 and tested on CIFAR-100 and SVHN in Figure 3 (a) and tested on CIFAR-100 and CelebA in Figure 3 (b). In Figure 3 (a), the

Table 1: Computational costs for Hessian eigenvalue decomposition

| Data size | $1\times16\times16$ | $1\times32\times32$ | $3\times32\times32$ |
|---|---|---|---|
| Number of parameters | 504,904 | 1,074,760 | 1,337,928 |
| Time cost | 1,414 s | 2,543 s | 3,371 s |
| Time/Parameter | (0.0028 s/dim) | (0.0023 s/dim) | (0.0025 s/dim) |

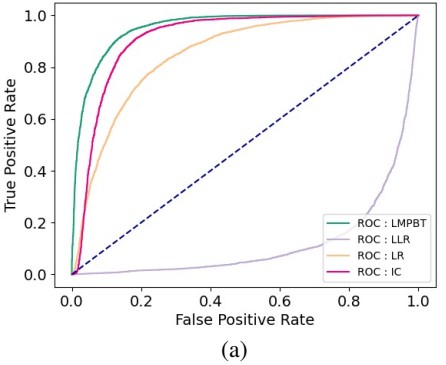 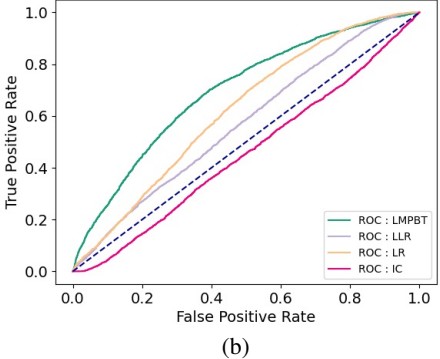

       (a)                                                  (b)

Figure 3: Comparison of ROC curves obtained using the LMPBT, LLR, IC, and LR for OOD detection with a VAE trained on the CIFAR-100 training dataset and (a) tested on the CIFAR-100 test dataset as in-distribution samples and SVHN test dataset as OOD samples and (b) tested on the CIFAR-100 test dataset as in-distribution samples and CelebA test dataset as OOD samples.

LBPBT (green line) shows higher true positive rates at all false positive rates than all other methods, including the IC (red purple line), which previously achieved state-of-the-art performance [18]. That is, the LMPBT universally outperformed the IC, regardless of the threshold for OOD detection. In contrast, the LLR (violet line) shows inferior performance compared to random guessing (blue line). This confirms that the LLR is not suitable for OOD detection when a VAE is used [18]. Similarly, in Figure 3 (b) using the CelebA dataset, the LMPBT achieved superior OOD detection performance compared to all other methods. All scores showed inferior performance compared with the case using the SVHN dataset. However, the LMPBT still outperformed the other methods even in this difficult OOD task.

For an in-depth evaluation of the LMPBT, we measured its performance using the AUROC and compared it with those of the competing methods, LLR, IC, and LR. We report the average AUROC over five repeated experiments using different random seeds. In all the experiments, the standard error of the AUROC was less than 0.001. A higher AUROC value indicates better performance. Tables 2 (a) and 2 (b) present the results for VAEs trained on CIFAR-10 and CIFAR-100, respectively, and tested on various datasets, including Noise, Constant, FMNIST, SVHN, MNIST, and CelebA. We can observe that the LMPBT achieved state-of-the-art performance in all cases. In particular, in the case of the model trained on CIFAR-10 and tested on SVHN, the LMPBT outperformed (AUROC: 0.931) the IC (AUROC: 0.912), which previously exhibited state-of-the-art performance [18]. Even for the CelebA test dataset, a more complicated dataset, the LMPBT using the model trained on CIFAR-10 showed superior performance (AUROC: 0.783) to that of the LR (AUROC: 0.714), which previously exhibited state-of-the-art performance [18]. Similarly, the LMBPT exhibited significantly superior performance when the model was trained on CIFAR-100, a more complicated dataset. For example, the LMPBT using the model trained on CIFAR-100 exhibited superior performance (AUROC: 0.683) to that of the LR (AUROC: 0.598).

We also present the results evaluated with AUPR and FPR80 in Supplementary Material. Further, we present the results using FMNIST as a training dataset with various test datasets considered in Table 2 in Supplementary Material. Additionally, we repeated the entire set of experiments using the

Table 2: Average AUROC of OOD detection methods on various test datasets over five repeated experiments using different random seeds.

| (a) Trained on CIFAR-10 | | | | | (b) Trained on CIFAR-100 | | | | |
|---|---|---|---|---|---|---|---|---|---|
| Dataset | LMPBT | LLR | IC | LR | Dataset | LMPBT | LLR | IC | LR |
| Noise | **1.0** | 1 | 0.032 | 0.994 | Noise | **1** | 1 | 0.993 | 0.993 |
| Constant | **1.0** | 0.258 | 1 | 0.974 | Constant | **1** | 0.042 | 0.999 | 0.948 |
| FMNIST | **0.994** | 0.074 | 0.992 | 0.991 | FMNIST | **0.997** | 0.325 | 0.987 | 0.970 |
| SVHN | **0.931** | 0.193 | 0.912 | 0.875 | SVHN | **0.956** | 0.123 | 0.912 | 0.820 |
| MNIST | **0.998** | 0.008 | 0.994 | 0.998 | MNIST | **0.998** | 0.376 | 0.979 | 0.995 |
| CelebA | **0.783** | 0.465 | 0.641 | 0.714 | CelebA | **0.683** | 0.575 | 0.464 | 0.598 |

Glow, and present the results in Supplementary Material. In summary, the results in Supplementary Material show that the LMPBT consistently outperformed across different metrics and datasets.

## 6 Conclusion

In this paper, we proposed a new OOD score for deep generative models based on a Bayesian hypothesis test. Specifically, we proposed the LMPBT for maximizing the probability that the alternative hypothesis is accepted when it is true among all alternative hypotheses that have the model parameters in the neighborhood of the model parameter specified under the null hypothesis. The LMPBT effectively performed OOD detection on all the tasks evaluated. We also addressed practical computational issues in the implementation of the LMPBT for deep generative models. For future research, more efficient computation of the LMPBT could be investigated; for example, recent second-order optimization methods could be used to compute the inverse of the Hessian matrix or its approximation. The proposed method can support reliable decision making by automatically detecting anomalies in complex systems in various real-world problems, such as manufacturing systems monitoring, fraud detection, and disease surveillance. However, there is also a potential risk that the proposed method could be used as a tool to identify and discriminate against minorities. To prevent this, we must ensure that the proposed method cannot be used for any purpose that could have negative social impacts.

## Acknowledgements

This work was supported by Samsung Electronics Co., Ltd (IO201209-07871-01) and the National Research Foundation of Korea (NRF) grant funded by the Korea Government (MSIT) (2018R1C1B6004511).

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
