# Supplementary Material for the Paper entitled "Locally Most Powerful Bayesian Test for Out-of-Distribution Detection Using Deep Generative Models"

Keunseo Kim[a], Juncheol Shin[a], and Heeyoung Kim[a,+]

[a]Department of Industrial and Systems Engineering, Korea Advanced Institute of Science and Technology (KAIST), Daejeon, Republic of Korea

[+]Corresponding author, Email: heeyoungkim@kaist.ac.kr

## A    Implementation Details

We present the implementation details for the VAE and Glow, used in Section 5. First, the structure of the Glow is presented in Table 1, where the "Level" refers to the number of scales that split the dimension of the latent space defined in the multi-scale architecture (Dinh et al., 2016), the "Depth per level" refers to the number of flow layers that are repeated on each scale, "In-channels × hidden units" refers to the number of input channels and the number of hidden units of the neural network that defines the parameters of the flow layers, and the "Coupling" refers to the type of the coupling layers. For the training, we used the Adam optimizer with learning rate of 0.001 and batch size of 64 for both datasets.

Next, Table 2 presents the structure for the VAE, which is the same as that used in Xiao et al. (2020). In Table 2, "nc" denotes the number of input channels, "nf" denotes the number of output channels with respect to the first convolutional neural network of the encoder, and "nz" denotes the dimension of the latent variable $\mathbf{z}$. Batch normalization layer (BN) and activation layer by ReLU function (ReLU) were added after each convolutional layer. We specified the hyperparameters as nc

Table 1: Structure for Glow

| Dataset | Level (L) | Depth per level (K) | In-channels × hidden units | Coupling |
|---------|-----------|---------------------|----------------------------|----------|
| FMNIST | 3 | 16 | $1 \times 512$ | Affine |
| CIFAR-10 | 3 | 16 | $3 \times 512$ | Affine |
| CIFAR-100 | 3 | 16 | $3 \times 512$ | Affine |

$= 1$, nf $=32$, and nz $=100$ for the FMNIST dataset; nc $= 3$, nf $=32$, and nz $=100$ for the CIFAR-10 and CIFAR-100 datasets. For training the VAE, we used the Adam optimizer with a learning rate of 0.0005, weight decay of 0.00003, and batch size of 64 for both datasets.

Table 2: Structure for VAE

| Encoder | Decoder |
|---------|---------|
| Input $\mathbf{x}$ | Input $\mathbf{z}$ reshape to nz $\times 1 \times 1$ |
| $4 \times 4$ Conv$_{nf}$ Stride 2, BN ReLU | $4 \times 4$ Deconv$_{4 \times nf}$ Stride 1, BN, ReLU |
| $4 \times 4$ Conv$_{2 \times nf}$ Stride 2, BN, ReLU | $4 \times 4$ Deconv$_{2 \times nf}$ Stride 2, BN, ReLU |
| $4 \times 4$ Conv$_{4 \times nf}$ Stride 2, BN, ReLU | $4 \times 4$ Deconv$_{nf}$ Stride 2, BN, ReLU |
| $4 \times 4$ Conv$_{2 \times nz}$ Stride 1 | $4 \times 4$ Conv$_{256 \times nc}$ Stride 2 |

# B  Additional Experimental Results

## B.1  Results of VAEs trained on CIFAR-10 and CIFAR-100 : AUPR and FPR80

We measured the performance of the LMPBT trained on CIFAR-10 and CIFAR-100 using the AUPR and FPR80, and compared it with those of the competing methods, LLR, IC, and LR in Tables 3-6, where ↑ represents that a higher value is better and ↓ represents that a lower value is better. As shown in the tables, the LMPBT consistently outperformed the other competing methods across different metrics and different datasets.

Table 3: AUPR↑ of OOD detection methods on various test datasets using VAEs trained on CIFAR-10

| Test dataset | LMPBT | LLR | IC | LR |
|---|---|---|---|---|
| Noise | **1** | **1** | 0.318 | 0.940 |
| Constant | **1** | 0.470 | 0.960 | 0.974 |
| FMNIST | **0.994** | 0.489 | 0.985 | 0.991 |
| SVHN | **0.941** | 0.337 | 0.922 | 0.841 |
| MNIST | **0.997** | 0.665 | 0.996 | 0.991 |
| CelebA | **0.791** | 0.511 | 0.665 | 0.724 |

Table 4: AUPR↑ of OOD detection methods on various test datasets using VAEs trained on CIFAR-100

| Test dataset | LMPBT | LLR | IC | LR |
|---|---|---|---|---|
| Noise | **1** | 0.976 | **1** | 0.996 |
| Constant | **1** | 0.965 | **1** | 0.980 |
| FMNIST | **0.928** | 0.911 | 0.910 | 0.844 |
| SVHN | **0.960** | 0.911 | 0.910 | 0.844 |
| MNIST | 0.996 | 0.854 | **0.998** | 0.948 |
| CelebA | **0.794** | 0.505 | 0.680 | 0.714 |

Table 5: FPR80↓ of OOD detection methods on various test datasets using VAEs trained on CIFAR-10

| Test dataset | LMPBT | LLR | IC | LR |
|---|---|---|---|---|
| Noise | **0** | **0** | 0.99 | 0.02 |
| Constant | **0** | 0.65 | **0** | 0.03 |
| FMNIST | **0.01** | 0.85 | 0.02 | 0.02 |
| SVHN | **0.04** | 0.91 | 0.05 | 0.18 |
| MNIST | **0.01** | 0.99 | 0.02 | 0.02 |
| CelebA | **0.53** | 0.60 | 0.76 | 0.59 |

## B.2 Results of Glow trained on CIFAR-10 and CIFAR-100 : AUROC, AUPR and FPR80

We measured the performance of the LMPBT trained on CIFAR-10 and CIFAR-100 using the AUPR and FPR80, and compared it with those of the competing methods, LLR, IC, and LR in

Table 6: FPR80↓ of OOD detection methods on various test datasets using VAEs trained on CIFAR-100

| Test dataset | LMPBT | LLR | IC | LR |
|---|---|---|---|---|
| Noise | **0** | 0.02 | 0.01 | 0.01 |
| Constant | **0** | 0.89 | **0** | 0.05 |
| FMNIST | **0.01** | 0.65 | 0.02 | 0.03 |
| SVHN | **0.07** | 0.89 | 0.09 | 0.22 |
| MNIST | **0.01** | 0.71 | 0.03 | 0.03 |
| CelebA | **0.55** | 0.60 | 0.78 | 0.58 |

Tables 7-12, where ↑ represents that a higher value is better and ↓ represents that a lower value is better. As shown in the tables, the LMPBT consistently outperformed the other competing methods across different metrics and different datasets.

Table 7: AUROC↑ of OOD detection methods on various test datasets using Glows trained on CIFAR-10

| Test dataset | LMPBT | LLR | IC | LR |
|---|---|---|---|---|
| Noise | **1** | 0.988 | **1** | 0.990 |
| Constant | **1** | 0.970 | **1** | 0.974 |
| FMNIST | **0.998** | 0.665 | 0.996 | 0.991 |
| SVHN | **0.967** | 0.932 | 0.928 | 0.839 |
| MNIST | **0.998** | 0.878 | **0.998** | 0.971 |
| CelebA | **0.786** | 0.690 | 0.743 | 0.531 |

Table 8: AUROC↑ of OOD detection methods on various test datasets using Glows trained on CIFAR-100

| Test dataset | LMPBT | LLR | IC | LR |
|---|---|---|---|---|
| Noise | **1** | 1 | 0.991 | 0.993 |
| Constant | **1** | 0.942 | 0.999 | 0.948 |
| FMNIST | **0.997** | 0.925 | 0.987 | 0.970 |
| SVHN | **0.941** | 0.823 | 0.912 | 0.820 |
| MNIST | **0.997** | 0.876 | 0.979 | 0.995 |
| CelebA | **0.731** | 0.575 | 0.464 | 0.598 |

Table 9: AUPR↑ of OOD detection methods on various test datasets using Glows trained on CIFAR-10

| Test dataset | LMPBT | LLR | IC | LR |
|---|---|---|---|---|
| Noise | **1** | **1** | 1.0 | 0.996 |
| Constant | **1** | 0.973 | 0.999 | 0.991 |
| FMNIST | **0.998** | 0.943 | 0.996 | 0.988 |
| SVHN | **0.963** | 0.846 | 0.945 | 0.857 |
| MNIST | **0.998** | 0.908 | 0.991 | 0.996 |
| CelebA | **0.743** | 0.685 | 0.494 | 0.671 |

Table 10: AUPR↑ of OOD detection methods on various test datasets using Glows trained on CIFAR-100

| Test dataset | LMPBT | LLR | IC | LR |
|---|---|---|---|---|
| Noise | **1** | **1** | 1.0 | 0.996 |
| Constant | **1** | 0.943 | 0.999 | 0.944 |
| FMNIST | **0.997** | 0.943 | 0.996 | 0.975 |
| SVHN | **0.941** | 0.846 | 0.932 | 0.845 |
| MNIST | **0.997** | 0.891 | 0.988 | 0.996 |
| CelebA | **0.722** | 0.671 | 0.474 | 0.641 |

Table 11: FPR80↓ of OOD detection methods on various test datasets using Glows trained on CIFAR-10

| Test dataset | LMPBT | LLR | IC | LR |
|---|---|---|---|---|
| Noise | **0** | **0** | 0.04 | 0.02 |
| Constant | **0** | 0.01 | **0** | 0.03 |
| FMNIST | **0.01** | 0.08 | **0.01** | 0.03 |
| SVHN | **0.03** | 0.28 | 0.05 | 0.18 |
| MNIST | **0.01** | 0.20 | **0.01** | 0.02 |
| CelebA | **0.34** | 0.64 | 0.81 | 0.58 |

## B.3   Results of the models trained on FMNIST: AUROC, AUPR, and FPR80

Similar to the left panel of Figure 2 in the main paper, we visualize the OOD detection ability of the LMPBT in Figure 1, which shows the log-scaled histogram of the LMPBT scores when FMNIST was used as an in-distribution dataset and MNIST was used as an OOD dataset. Figures 1(a) and

Table 12: FPR80↓ of OOD detection methods on various test datasets using Glows trained on CIFAR-100

| Test dataset | LMPBT | LLR | IC | LR |
|:---:|:---:|:---:|:---:|:---:|
| Noise | **0** | **0** | 0.99 | 0 |
| Constant | **0** | 0.04 | 0 | 0.05 |
| FMNIST | **0.01** | 0.10 | **0.01** | 0.04 |
| SVHN | **0.04** | 0.31 | 0.07 | 0.28 |
| MNIST | **0.01** | 0.23 | **0.01** | 0.05 |
| CelebA | **0.32** | 0.68 | 0.83 | 0.61 |

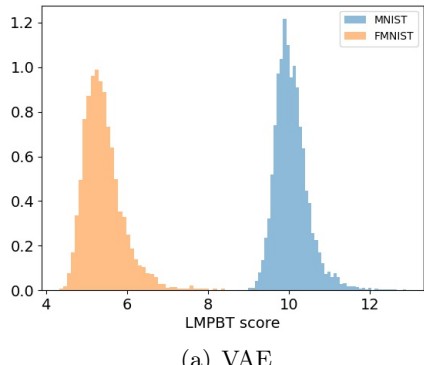

(a) VAE

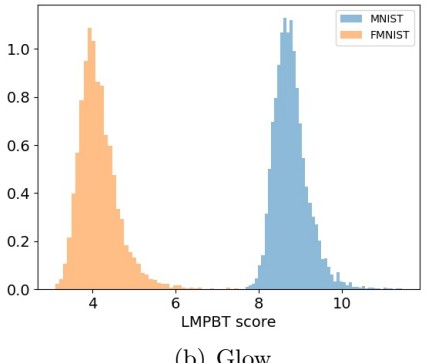

(b) Glow

Figure 1: (a) Histogram of the LMPBT scores obtained with a VAE trained on FMNIST and tested on FMNIST and MNIST . (b) Histogram of the LMPBT scores obtained with a Glow trained on FMNIST and tested on FMNIST and MNIST.

1(b) present the results obtained with a VAE and a Glow, respectively. As depicted in Figure 1, both VAE and Glow assigned significantly higher LMPBT scores to the OOD samples. It is notable that the OOD samples were more clearly distinguished from the in-distribution samples in this FMNIST experiments than the CIFAR-10 experiments in Figure 2 (left) in the main paper.

We also compared the ROC curve of using the LMPBT with those of using the LR, IC, and LLR in OOD detection using a VAE and a Glow trained on FMNIST and tested on FMNIST and MNIST. As shown in Figure 2, the LMPBT (green line) perfectly detected OOD samples for both cases of the VAE and Glow. In contrast, the IC (red purple line) and LR (orange line) showed unreliable performance when Glow was used as shown in Figure 2(b).

We measured the performance of the LMPBT using the AUROC, AUPR, and FPR80, and

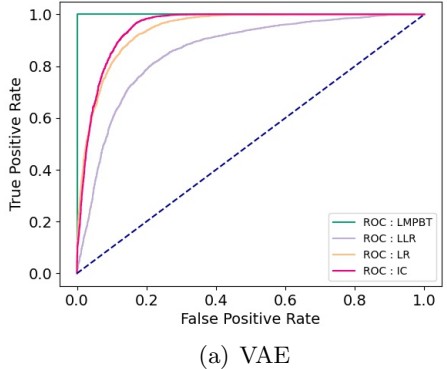 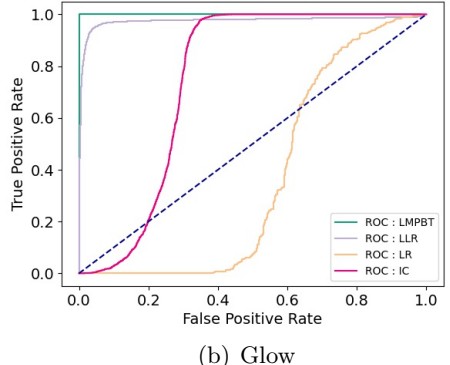

(a) VAE                              (b) Glow

Figure 2: Comparison of ROC curves of using the LMPBT, LLR, IC, and LR for OOD detection using (a) a VAE and (b) a Glow trained on FMNIST training dataset and tested on FMNIST test dataset and MNIST test dataset.

compared it with those of the competing methods, LLR, IC, and, LR in Table 13 to Table 18. We can see that the LMPBT showed almost perfect performance on both VAE and Glow in terms of all three different metrics on all the test datasets. In contrast, the competing methods showed unreliable performance in certain cases, similar to the CIFAR-10 experiments. For example, the LLR showed inferior performance (AUROC: 0.688) when tested on Constant dataset using VAE. Similarly, the IC showed inferior performance (AUROC: 0.455) when tested on Noise dataset using VAE, and the LR showed inferior performance (AUROC: 0.629) when tested on MNIST dataset using Glow.

Table 13: AUROC↑ of OOD detection methods on various test datasets using VAEs trained on FMNIST

| Test dataset | LMPBT | LLR | IC | LR |
|:---:|:---:|:---:|:---:|:---:|
| MNIST | **1** | 0.858 | 0.951 | 0.942 |
| SVHN | **1** | 0.698 | 0.999 | 0.999 |
| CIFAR-10 | **1** | 0.950 | 0.936 | 0.995 |
| Noise | **1** | **1** | 0.455 | 0.988 |
| Constant | **1** | 0.688 | **1** | 0.996 |

Table 14: AUROC↑ of OOD detection methods on various test datasets using Glows trained on FMNIST

| Test dataset | LMPBT | LLR | IC | LR |
|---|---|---|---|---|
| MNIST | **0.998** | 0.975 | 0.747 | 0.629 |
| SVHN | **1** | **1** | 0.992 | 0.993 |
| CIFAR-10 | **1** | **1** | 0.991 | 0.993 |
| Noise | **1** | **1** | **1** | **1** |
| Constant | **1** | **1** | 0.996 | 0.995 |

Table 15: AUPR↑ of OOD detection methods on various test datasets using VAEs trained on FMNIST

| Test dataset | LMPBT | LLR | IC | LR |
|---|---|---|---|---|
| MNIST | **1** | 0.860 | 0.964 | 0.952 |
| SVHN | **1** | 0.566 | 0.999 | 0.999 |
| CIFAR-10 | **0.998** | 0.906 | 0.942 | 0.995 |
| Noise | **1** | **1** | 0.662 | 0.992 |
| Constant | **1** | 0.553 | **1** | 0.995 |

Table 16: AUPR↑ of OOD detection methods on various test datasets using Glows trained on FMNIST

| Test dataset | LMPBT | LLR | IC | LR |
|---|---|---|---|---|
| MNIST | **0.994** | 0.942 | 0.849 | 0.758 |
| SVHN | **1** | **1** | 0.995 | 0.996 |
| CIFAR-10 | **1** | **1** | 0.993 | 0.995 |
| Noise | **1** | **1** | **1** | **1** |
| Constant | **1** | **1** | 0.997 | 0.997 |

Table 17: FPR80↓ of OOD detection methods on various test datasets using VAEs trained on FMNIST

| Test dataset | LMPBT | LLR | IC | LR |
|---|---|---|---|---|
| MNIST | **0** | 0.210 | 0.081 | 0.096 |
| SVHN | **0** | 0.892 | **0** | 0.000 |
| CIFAR-10 | **0** | 0.014 | 0.120 | 0.003 |
| Noise | **0** | **0** | 0.580 | 0.017 |
| Constant | **0** | 0.986 | **0** | 0.000 |

Table 18: FPR80↓ of OOD detection methods on various test datasets using Glows trained on FMNIST

| Test dataset | LMPBT | LLR | IC | LR |
|--------------|-------|-----|-----|-----|
| MNIST | **0** | 0.011 | 0.304 | 0.462 |
| SVHN | **0** | **0** | 0.009 | 0.006 |
| CIFAR-10 | **0** | **0** | 0.010 | 0.006 |
| Noise | **0** | **0** | **0** | **0** |
| Constant | **0** | **0** | 0.007 | 0.004 |