# OpenReview forum: "Locally Most Powerful Bayesian Test for Out-of-Distribution Detection using Deep Generative Models"
_NeurIPS.cc/2021/Conference — NeurIPS 2021 Poster_

### Official Review · Reviewer_6JoX · 2021-07-16

**Rating:** 7
**Confidence:** 4

**Summary:**

This paper proposes a new method for out-of-distribution (OOD) detection using deep generative models. In particular, the paper leverages a Bayesian hypothesis testing framework, and is based the so-called „locally most powerful Bayesian test“, which enjoys certain theoretical guarantees (namely that it maximizes the probability of correct OOD detection) under some conditions. It is described how the intractable test can be efficiently implemented in practice (when used in conjunction with deep generative models), leveraging low-rank approximations of the involved Hessian matrix. Empirical evaluations demonstrate that the proposed approach compares favorably to alternative methods in a variety of settings, for both VAE and Glow models.

**Limitations And Societal Impact:**

The paper describes its limitations only to some degree. As pointed out above, the computational complexity of the approach is not assessed sufficiently. Without an analysis of the compute-performance trade-off (of both this method and competing baselines), it is difficult to assess the significance of the method. Potential negative societal impact is addressed sufficiently.

**Main Review:**

**Post Rebuttal**

The author response convincingly addressed the main concerns I had raised -- I therefore increase my score from 6 (weak acceptance) to 7 (acceptance).

---

This manuscript proposes a new method to tackle the important OOD detection problem. It is therefore relevant to the NeurIPS community and has potentially high impact. The method is theoretically motivated by Bayesian hypothesis testing, is intuitively sensible and easy-to-understand. The experiments clearly demonstrate that the proposed approach performs strongly compared to other recently-developed OOD detection methods for deep generative models. However, the computational complexity of the method is not entirely transparent. The required Hessian approximation seems fairly expensive, which is only insufficiently analyzed. (See comments below for more details.) Adding such an analysis to the paper would significantly strengthen it. All in all, I am tending towards recommending acceptance of the paper — I am happy to strengthen my recommendation if the authors can convincingly address the raised issues.


Detailed comments:
- I like the idea of explicitly formulating the OOD detection as a Bayesian hypothesis test, which to my knowledge has not been done in prior work.
- The proposed approach is conceptually simple, intuitively sensible and appears to be theoretically grounded and sound. The paper also provides some useful theoretical analysis of the properties of the Bayesian test used for OOD detection.
- It is described how the (originally intractable) method can be implemented in practice by doing a low-rank approximation of the required full-network Hessian (which is implemented using power iteration) and using Hessian-vector products that avoid computation and storage of the full Hessian. While this approach seems reasonable, it is not clear how the computational complexity compares to that of other OOD detection methods; using power iteration to obtain the top eigenvectors/-values is expected to still be comparably expensive, so a more thorough and transparent analysis of the computational cost would improve the paper significantly (see also below for a the comment on the short analysis provided in the experiments).
- They describe how their Bayesian testing framework generalizes previous likelihood-ratio based OOD detection methods, which is insightful.
- The empirical evaluation convincingly demonstrates the effectiveness of the proposed approach in a variety of settings, as compared to strong baselines such as LR, LLR and IC, for different types of deep generative models (in particular VAEs and Glow).
- They study how the rank of the low-rank approximation of the Hessian (i.e. the number of eigenvalues considered) affects the OOD detection performance, showing that even for deep neural network models, a small number of eigenvalues (i.e. 20-30) achieves the best performance. However, Fig 2 (right) is not entirely conclusive in that the performance seems to degrade for a larger number of estimated eigenvalues (i.e. a low-rank approximation with a higher rank). Clearly, in theory, performance should become better with more accurate Hessian estimation, so it would be great to know why this is not the case in practice, and I would appreciate the author’s comments on this. Is this simply result of numerical instabilities when using power iteration to estimate the top eigenvalues? If so, why is this not an issue in practice, and how would we choose the optimal number of eigenvalues in practice?
- They also analyze the computational cost of the proposed method, which seems comparable to LR. It would be great to also report the computational cost of LLR and IC, which are likely much cheaper as they only require likelihood computations. Otherwise it is somewhat difficult to assess the compute-performance trade-off of the proposed approach, which is necessary to fairly compare different methods. This is the main issue that I see with the described method.

**Time Spent Reviewing:**

5

---

> ### Author Response · Authors · 2021-08-10
> **Responses to your valuable comments**
>
> Thank you for your careful review of our paper and for the insightful and constructive comments. Please find our detailed answers to your comments below.
>
> ***<Responses to "Detailed comments">***
> **1. (Computational cost for Hessian eigenvalue decomposition)**
> To evaluate the computational cost for computing top $r$ Hessian eigenvectors,  we performed two sets of additional experiments.
>
> First, we compared the computational cost required for the Hessian eigenvalue decomposition with that for network parameter estimation, which can be considered as a baseline because network parameter estimation is equally performed for other OOD detection methods. In our additional experiments using FMNIST (1x32x32) with VAEs, it took 2543 s to obtain the top 30 eigenvectors. This cost is relatively small (approximately 18%) compared with the cost for network parameter estimation (13448 s).
>
> Second, we evaluated the scalability of Hessian eigenvalue decomposition. Specifically, we measured the computational cost for the eigenvalue decomposition of the Hessian (top 30 eigenvectors) for datasets of various sizes with VAEs. We also calculated the ratio of time cost to the number of parameters to see the relationship between the dimension of the Hessian and computational cost. The results are as in the following table. The ratio was obtained as 0.0028, 0.0023, and 0.0025 s for various data sizes. The ratio remained similar, which indicates that the computational cost tends to increase linearly with the number of parameters. These results show that our method is scalable and can be applicable to large-scale networks.
>
> | Data size            | 1x16x16        | 1x32x32        | 3x32x32        |   |
> |----------------------|----------------|----------------|----------------|---|
> | Number of parameters | 504904         | 1074760        | 1337928        |   |
> | Time cost            | 1414 s         | 2543 s         | 3371 s         |   |
> | Time/Parameter       | (0.0028 s/dim) | (0.0023 s/dim) | (0.0025 s/dim) |   |
>
> **2. (Explanation for the best performance with top 30 eigenvalues)**
> The behavior that the performance does not increase monotonically with the number of eigenvalues may be explained by an identifiability issue of overparameterized models. For overparameterized models, there are often  "degenerate” directions in the parameter space [1], which are corresponding to the smallest eigenvalues of the Hessian. The change in the parameters in degenerate directions leads to almost no difference in the outputs and causes an identifiability issue. Deep generative models also can suffer from this identifiability issue. However, we may overcome this issue by using a low-rank approximation of the Hessian with the largest eigenvalues. Because changes in the output values in the degenerate directions can be considered like random noise, the test accuracy of a compressed model can sometimes be better than that of the full model. According to [2], updating the parameters in the sharpest directions, which are corresponding to the top $r$ eigenvectors, sometimes improves the generalization ability. Specifically, the authors analyzed the dynamics of stochastic gradient descent (SGD) in the subspaces of the sharpest directions of the loss surface (i.e., directions of top $r$ eigenvectors of the Hessian), called Nudged-SGD (NSGD), and their influences on the training speed and the generalization ability of the final model.  The authors showed that the NSGD can optimize significantly faster and find good generalizing sharp minima, i.e., minima exhibiting better generalization performance in the loss surface, which are sharper compared to the minima found by baseline SGD. However, as discussed in [2], the structure of the Hessian is highly dependent on the dataset, and thus the impact of parameter updates in the sharpest directions on the final generalization can be different depending on the dataset.
>
> To further investigate whether the behavior that the performance does not increase monotonically with the number of eigenvalues was due to numerical instability when using power iteration, we performed additional experiments. We computed the top 30 eigenvalues ten times with different random seeds. The mean and standard deviation of the top three eigenvalues over the ten experiments ​​are summarized in the following table. Considering the relatively small standard deviations, it can be said that the eigenvalues are stably obtained.
>
> |             | mean        | standard deviation    |       |
> |:----------------------:|:----------------:|:----------------:|:----------------:|
> |  1st eigenvalue   |  $4.4 \times10^5$          |  $0.05 \times 10^5$         |   |
> | 2nd eigenvalue    | $3.2 \times 10^5$         | $0.18 \times 10^5$          |   |
> | 3rd eigenvalue     | $2.8 \times 10^5$         | $0.16 \times 10^5$  |   |
>
>
> **3. (Determination of the optimal number of eigenvalues in practice)**
> In our experiments, we determined the number of eigenvalues based on experiments using a validation dataset. Alternatively, we may use the ratio of the sum of the chosen top $r$ eigenvalues to the Hessian trace, similar to the way for choosing the number of principal components in principal component analysis (PCA) [3]. Specifically, let $Tr(H)$ denote the Hessian trace and $\lambda_j$ denote the top $j$th eigenvalue. Then, the explanatory power of the $j$th eigenvector can be expressed as $\lambda_j / Tr(H)$. We can set a threshold $p$ representing the desired level of explanatory power in advance (e.g., $p$=0.7), and choose the optimal value of $r$ as the smallest integer that satisfies $\sum_{j=1}^{r} \lambda_j / Tr (H) > p$. In our experiments, the ratio was calculated to be 0.68 with the top 30 eigenvalues. The Hessian trace can be easily calculated using the Hutchinson method  [4], which is supported in the Pyhessain package [5].
>
>
> **4. (Computational cost of LLR and IC & compute-performance trade-off)**
> As you suggested, we performed additional experiments and measured the computational cost of LLR and IC. The LLR and IC spent 0.0125 s and 0.0128 s to test a single sample, respectively, while the LMPBT (our method) and LR spent 0.11 s and 0.30 s to test a single sample, respectively. If we compare the four methods in terms of compute-performance trade-off, LMPBT and LR may be considered as methods sacrificing some computational cost for accuracy, whereas LLR and IC may be considered as those sacrificing some accuracy for computational cost. We will add the computational cost of LLR and IC, and discuss the compute-performance trade-off in the revised version of our paper.
>
> ***<Responses to "Limitations And Societal Impact">***
> In the final version, we will add the results of our additional experiments to assess the computational complexity of our method and discuss the compute-performance trade-off of our method and competing baselines, as we answered to your "Detailed comments" #1~#4 above.
>
> ***
> [1] Maddox, Wesley J., Gregory Benton, and Andrew Gordon Wilson. "Rethinking parameter counting in deep models: Effective dimensionality revisited." arXiv preprint arXiv:2003.02139 (2020).
> [2] Jastrzebski, Stanislaw, et al. "On the Relation Between the Sharpest Directions of DNN Loss and the SGD Step Length." ICLR (2019).
> [3] Abdi, Hervé, and Lynne J. Williams. "Principal component analysis." Wiley interdisciplinary reviews: computational statistics 2.4 (2010): 433-459.
> [4] Avron, Haim, and Sivan Toledo. "Randomized algorithms for estimating the trace of an implicit symmetric positive semi-definite matrix." Journal of the ACM (JACM) 58.2 (2011): 1-34.
> [5] Yao, Zhewei, et al. "Pyhessian: Neural networks through the lens of the hessian." 2020 IEEE International Conference on Big Data (Big Data). IEEE, 2020.

---

> > ### Comment · Reviewer_6JoX · 2021-08-24
> > **Updated review to recommend acceptance**
> >
> > Thanks a lot for the detailed and insightful response!
> > This clarifies my main concerns -- I therefore do not see a reason to hold back my score and will update my review to recommend acceptance (7).

---

> > > ### Author Response · Authors · 2021-08-24
> > > **Thanks for your response!**
> > >
> > > Thank you for your additional efforts and time to read our response. We are happy to hear that our response clarified your concerns. We sincerely appreciate your insightful comments that helped us improve the manuscript.

---

### Official Review · Reviewer_Huvy · 2021-07-16

**Rating:** 6
**Confidence:** 5

**Summary:**

**Summary**
This paper presents a novel OOD detection method for deep generative models. The key idea is to formulate OOD detection as a Bayesian hypothesis test, where the OOD score is the log of likelihood ratio under the original network parameters $\theta_0$ and new parameters $\theta_1$ when adding the test sample $\mathbf{x}_t$. The overall testing framework is similar to previous works on likelihood ratio (Ren et al. 2019) and likelihood regret (Xiao et al. 2020). The novelty lies in the estimation of $\theta_1$ under the alternative hypothesis. Specifically, the authors estimate new network parameters using the second-order Hessian information. The authors also propose strategies to reduce the computational cost. Evaluations on both CIFAR-10 and CIFAR-100 demonstrate superior performance.

**Ethics Review Area:**

["I don’t know"]

**Limitations And Societal Impact:**

The author does not include on in the main paper but this doesn't affect my scoring.

**Main Review:**

**Strengths**:
- Authors extensively evaluate the proposed method under two generative models (VAE and Glow) and various datasets.
- The performance improvement over the prior state-of-the-art, name likelihood regret, is significant.
- The test is mathematically grounded.
- The authors spent efforts addressing two key challenges related to the computational issues in Section 4: (1) low-rank approximation of the hessian and (2) inverse of Hessian. The experimental results in L268-L274 also presented convincing evidence.
- The writing is mostly clear but can be improved.

**Weaknesses**:

- Likelihood Regret [r1] seems to share the same spirit in terms of estimating the new network parameters when the test sample x_t is added to the training set.  [r1] finetunes the model with the test sample for a few iterations, which seems to be the upper bound of the authors' algorithm., i.e, the proposed method is the approximation of the fine-tuned parameters. An explanation and discussion about why the authors' method surpasses  [r1] are needed.
- According to the authors' code, they need a preprocessing step to compute the top-k Hessian eigenvectors and eigenvalues of the original model using power iteration across the whole training set. The computation budget would be very high given this pre-processing approach. More discussion on this would be appreciated.
- The computational overhead could be significantly higher when employing the proposed method on the high-resolution datasets and large models due to the computation of Hessian. While the reviewer understands it's a general concern shared by second-order methods, it'd be informative to test the limits of the approach.

Minor:
Typo in the caption of Table 6 in Appendix, it should be the results under CIFAR-100, not CIFAR-10.

[r1] Z. Xiao, Q. Yan, and Y. Amit. Likelihood regret: An out-of-distribution detection score for variational auto-encoder. arXiv preprint arXiv:2003.02977, 2020.

**Time Spent Reviewing:**

3

---

> ### Author Response · Authors · 2021-08-10
> **Responses to your valuable comments**
>
> Thank you for your careful review of our paper and for the insightful and constructive comments. Please find our detailed answers to your comments below.
>
> ***<Responses to "Weakness">***
> **1. (Comparison with the LR)**
> Likelihood Regret (LR) is closely related to our method, but there are significant differences between the LR and our method as follows:
> Our method updates new parameters using the expanded dataset in which the test sample and training dataset are combined. However, in the case of LR, new parameters are updated using only the test sample.
> Instead, LR updates the parameters for only a few iterations such that the updated parameters do not differ significantly from MLE. This learning process may be unstable depending on the number of iterations and required additional regularizations. For example, Xiao et al.  [1] showed poor performance of the LR when both encoder and decoder parameters of VAEs were updated (Table 3 of [1]); therefore, they fixed the encoder parameters and updated only the decoder parameters for the regularization.
> In contrast, our method does not require additional regularizations, because we update the parameters using an expanded dataset containing the training dataset.
> Furthermore, both LR and our method can be interpreted as a hypothesis test for OOD detection using $\theta_1$ that satisfies the neighborhood condition, but with different values of $\theta_1$. Because our method is locally most powerful according to Proposition 1, the OOD detection performance of our method is guaranteed to be superior to LR.
>
> **2. (Computational cost for top-k Hessian eigenvectors and eigenvalues)**
> As you mentioned, our method requires the computation of top-k Hessian eigenvectors using the whole training dataset.  In our experiments using FMNIST (1x32x32) with VAEs, it took 2543 s to obtain the top-30 eigenvectors. This cost is relatively small (approximately 18%) compared with the cost for training the model (13448 s). We will add this analysis of additional cost due to the computation of top-k Hessian eigenvectors in the final version.
>
> **3. (Computational cost for large sized data)**
> As you mentioned, the exact eigenvalue decomposition of the Hessian requires a huge computational cost, $O(n^3)$. However, the computational cost for our method does not significantly increase even if the data size increases. Specifically, we used power iteration for the eigenvalue decomposition of the Hessian. The power iteration can be performed without explicitly computing the full Hessian matrix [2]; instead, it is performed using the Hessian-vector products, which can be computed efficiently with $O(n)$ [3]. In our study, we evaluated the computational cost for the eigenvalue decomposition of the Hessian (top 30 eigenvectors) for datasets of various sizes with VAEs. We also calculated the ratio of time cost to the number of parameters to see the relationship between the dimension of the Hessian and computational cost. The results are as in the following table. The ratio was obtained as 0.0028, 0.0023, and 0.0025 s for various data sizes. The ratio remained similar, which indicates that the computational cost tends to increase linearly with the number of parameters. This is consistent with the theoretical analysis of the cost in [3].
>
> | Data size            | 1x16x16        | 1x32x32        | 3x32x32        |   |
> |----------------------|----------------|----------------|----------------|---|
> | Number of parameters | 504904         | 1074760        | 1337928        |   |
> | Time cost            | 1414 s         | 2543 s         | 3371 s         |   |
> | Time/Parameter       | (0.0028 s/dim) | (0.0023 s/dim) | (0.0025 s/dim) |   |
>
> ***<Responses to "Minor">***
> Sorry for the typo. We will correct the typo in the caption of Table 6 in Appendix.
> ***
> [1] Xiao, Zhisheng, Qing Yan, and Yali Amit. "Likelihood Regret: An Out-of-Distribution Detection Score For Variational Auto-encoder." Advances in Neural Information Processing Systems 33 (2020).
> [2] Yao, Zhewei, et al. "Pyhessian: Neural networks through the lens of the hessian." 2020 IEEE International Conference on Big Data (Big Data). IEEE, 2020.
> [3] Pearlmutter, Barak A. "Fast exact multiplication by the Hessian." Neural computation 6.1 (1994): 147-160.

---

### Official Review · Reviewer_ossS · 2021-07-17

**Rating:** 6
**Confidence:** 3

**Summary:**

The paper proposes a new Bayesian hypothesis test for out-of-distribution detection (OOD) based on a trained generative model. The problem is formulated as a hypothesis testing on the parameter of the model where the null hypothesis $H_0$ posits that the model parameter is $\theta_0$ and the alterternative $H_1$ states that the parameter is $\theta_1$. The test statistic is the ratio of the posterior probabilities of the two (Bayes factor. See (5)). This way to formulate the problem is common and $\theta_0$ is commonly set to the maximum likelihood estimate (MLE) on the training data (in-distribution samples).

The main contribution of this paper is in setting $\theta_1$. The paper proposes to set it by MLE but on data constructed by combining the training data and the test point to test (i.e., so one more point added to the training data. See (7)). The paper shows that doing this leads to a test that satisfies a new notion called “locally most powerful Bayesian test” (LMPBT) given in Def 3.1. Since fully estimating the MLE on the augmented training data for each point to test is expensive, the paper further proposes a way to scale up, based on the use of the upweighting method from [9], and Hessian approximation (see (8) and Sec 4.1). The resulting MLE solution on the augmented data can be written as an additive perturbation (involving a low-rank Hessian) of the MLE estimate from the original training data. In this way, the MLE on the training data can be estimated only once, and reused for each OOD test point. In experiments, the new Bayesian test is shown to outperform competing approaches on standard benchmark datasets such as CIFAR10, CIFAR100, SVHN, MNIST, and Fashion MNIST.


**Ethical Concerns:**


No concerns


**Limitations And Societal Impact:**

yes

**Main Review:**


*Clarity*:

The paper is fairly well written. The problem setting and background are briefly given just enough to understand the contribution, which is good. Preciseness in mathematical statements can be improved. For instance, the use of $\approx$ in the proof of Prop 1 makes it unclear about the effect of the omission of the remainder term in the Taylor expansion. It is also unclear what “sufficiently small $\delta$” means in Def 3.1 (i.e., how small is small). Other than these, the main ideas presented and the details of the proposed method are easy to follow.

*Originality*:

The use of Bayes factor for OOD is natural and has been considered before. The main contributions are
1. Setting $\theta_1$ by MLE on the augmented training data (training data + the test point);
2. The use of upweighting ((9)) to avoid a full MLE computation for each point, as well as its further low-rank Hessian approximation (Sec 4.1) to scale up;
3. The result in Prop 4.1 stating that setting $\theta_1$ in this way results in a locally most powerful test.

The upweighting method is known from [9], and the low rank Hessian approximation is a commonly used operation especially in the optimization literature among others. In that sense, I see that the originality mainly comes from the idea in 1) and the result in 3). Overall I find that the paper has nicely put together these results and makes a practical approach.

*Quality*:

There is some impreciseness in the statement of Prop 4.1, and Def 3.1 however. More in the questions below. Up to these impreciseness, the proposed Bayesian approach appears to be technically sound.


*Significance*:

The proposed test gives strong empirical results, outperforming competing methods. Empirically, the approach is deemed significant, and may open up a new direction for OOD. There are some unclear points regarding experiments, as discussed below. Theoretical significance is lacking however. Besides showing that the approach is a locally most powerful Bayesian test (LMPBT), there is no further elaboration on the implications of this statement.  The definition of the LMPBT also appears to be proposed by the present work for the first time, tailored to fit the proposed approach (combining [6] and [12]).

## Questions
1. In the definition of LMPBT in Def 3.1, how small is small for $\delta$?
2. What are the implications of Proposition 1? That is, the test is a LMPBT. What can be said more about this? Is it the only test that satisfies this definition?
3. In (9), the Hessian-gradient product term carries a factor of 1/n where n is the sample size. What happens asymptotically (i.e., when n goes to infinity)? Does it mean $\tilde{\theta}_t$ (MLE on the augmented data) would be the same as $\hat{\theta}_t$ (MLE on the training data)? What happens to the decision of the test in that case?
4. In Fig 2b, why does the AUC curve fluctuate as $r$ the number of eigenvectors in approximating the Hessian increases? Naturally one would expect the approximation to get better as $r$ increases.
5. In Fig 3, what does “tested on CIFAR-100 test data and SVHN” mean? Are they combined into one test set? I’m trying to see why the likelihood ratio LLR shows completely different behaviors in  Fig 3a and Fig 3b.

—-


**Time Spent Reviewing:**

4

---

> ### Author Response · Authors · 2021-08-10
> **Responses to your valuable comments**
>
> Thank you for your careful review of our paper and for the insightful and constructive comments. Please find our detailed answers to your comments below.
>
> ***<Responses to “Clarity” and “Quality”>***
> We will improve the mathematical statements to be more precise by adding detailed explanations about the remainder term in the Taylor expansion and the meaning of “sufficiently small” $\delta$. Please see more details in our response to your Question #1 below.
>
> ***<Responses to "Significance">***
> - We will clarify unclear points regarding experiments, including the effect of the number of eigenvectors in the Hessian approximation and the behaviors of LLR in Fig 3. Please see more details in our responses to your Questions #4 and #5 below.
> - We will explain the implication of Proposition 1 in more detail. Please see more details in our response to your Question #2 below.
> - As you mentioned, we propose the locally most powerful Bayesian test (LMPBT) for the first time. We propose a Bayesian testing framework that generalizes previous likelihood-ratio based methods for OOD detection. We prove that our specification of the alternative hypothesis guarantees to maximize the probability of detection of OOD samples. We also confirm this via numerical experiments. Please see more details in our response to your Question #2 below.
>
> ***<Responses to "Questions">***
> **1. (How small is $\delta$ in Def 3.1?)**
> We assume a small $\delta$ to apply the second-order Taylor approximation in the proof for Proposition 1. In Equation (12), we approximate the log-Bayes factor using the second-order Taylor approximation, assuming that higher-order remainder term is negligible. In fact, the remainder term is upper-bounded by $\frac{M}{6}||\theta_1-\hat{\theta}||^3$ (Lagrange error bound), where $M$ is an upper bound of the third derivative of the log-likelihood for $||\theta_1-\hat{\theta}||<\delta$. With sufficiently small $\delta$, the first- and second-order terms will dominate the remainder term because higher order terms vanish much more rapidly, and the remainder term will be negligible. Therefore, the size of $\delta$ is related to the error bound for the second-order Taylor approximation, and the $\delta$ needs to be “small” such that the error bound $\frac{M}{6} ||\theta_1-\\hat{\theta}||^3$ is negligible for $||\theta_1-\hat{\theta}||<\delta$. We will clarify this in the final version.
>
> **2. (Implication of Proposition 1)**
> If a test is a LMPBT, the probability of correct detection of OOD sample is maximized. More specifically, if we formulate the OOD detection problem as a Bayesain hypothesis test with $H_0: \theta=\theta_0$ (a test sample is in-distribution) vs. $H_1: \theta=\theta_1$ (a test sample is OOD), the OOD detection performance depends the specification of $\theta_0$ and $\theta_1$, which are representative parameters for in-distribution and OOD samples, respectively. Typically,  $\theta_0$ is chosen as the MLE of the in-distribution training data. However, it is difficult to specify $\theta_1$ under the lack of information on OOD samples. In our study, we propose to choose $\theta_1$ as in Equation (9), because this choice of $\theta_1$ makes the test a LMPBT that maximizes the probability of correct detection of OOD samples. In fact, the previous likelihood-ratio based OOD detection methods can be expressed using the same form of Bayesian hypothesis test with different $\theta_1$ values. For example, the LLR [1] specified $\theta_1$ as the MLE of noise-perturbed input data, and the LR [2] specified $\theta_1$ as that updated from the MLE for a few iterations given a test sample.  In contrast, we specify $\theta_1$ according to Equation (9), based on a theoretical analysis that this choice of $\theta_1$ guarantees to maximize the probability of detection of OOD samples. This theoretical result is also empirically validated using various benchmark datasets in Section 5, where we show that the proposed method outperforms the previous methods.
>
> **3. (What if n goes to infinity in Eq. (9)?)**
> As you mentioned, $\tilde\theta_t$ asymptotically converges to $\hat\theta$. However, the convergence rate for $\tilde\theta_t$ is different depending on whether the test sample is an in-distribution or an OOD sample. If the test sample is an in-distribution sample, the value of the Hessian-gradient product will converge to 0 due to the asymptotic normality of the MLE. Formally,  $H^{-1}_{\hat{\theta}} {\nabla}_\theta(-\log L(\hat{\theta}|\{x}_t))$ converges to 0 with a convergence rate of $1/\sqrt{n}$. Then, the convergence rate of $\tilde{\theta}_t$ to $\hat{\theta}$ will be $1/{n^{1.5}}$. In contrast, if the test sample is OOD, $H_\hat\theta^{-1}\nabla_\theta(-\log L(\hat{\theta}|\{x}_t))$ will not converge to 0 because the loss gradient differs from 0. In this case, the convergence rate of $\tilde{\theta}_t$ to $\hat{\theta}$ will be $1/n$. Because the proposed method detects OOD samples based on a threshold calculated by the distribution of OOD scores obtained using the in-distribution samples, the different convergence rates enable the detection of the OOD samples even for very large sample cases. In fact, our method performed effectively in OOD detection tasks using CIFAR-10 experiments, which were based on a large number of samples (50000 samples).
>
> **4. (Why does the AUC curve fluctuate with $r$ in Fig 2b?)**
> The behavior that the performance does not increase monotonically with the number of eigenvalues may be explained by an identifiability issue of overparameterized models. For overparameterized models, there are often  "degenerate” directions in the parameter space [3], which are corresponding to the smallest eigenvalues of the Hessian. The change in the parameters in degenerate directions leads to almost no difference in the outputs and causes an identifiability issue. Deep generative models also can suffer from this identifiability issue. However, we may overcome this issue by using a low-rank approximation of the Hessian with the largest eigenvalues. Because changes in the output values in the degenerate directions can be considered like random noise, the test accuracy of a compressed model can sometimes be better than that of the full model. According to [4], updating the parameters in the sharpest directions, which are corresponding to the top $r$ eigenvectors, sometimes improves the generalization ability. Specifically, the authors analyzed the dynamics of stochastic gradient descent (SGD) in the subspaces of the sharpest directions of the loss surface (i.e., directions of top $r$ eigenvectors of the Hessian), called Nudged-SGD (NSGD), and their influences on the training speed and the generalization ability of the final model.  The authors showed that the NSGD can optimize significantly faster and find good generalizing sharp minima, i.e., minima exhibiting better generalization performance in the loss surface, which are sharper compared to the minima found by baseline SGD. However, as discussed in [4], the structure of the Hessian is highly dependent on the dataset, and thus the impact of parameter updates in the sharpest directions on the final generalization can be different depending on the dataset.
>
> **5. (The meaning of the caption in Fig 3 & behaviors of LR in Fig 3)**
> In Fig 3, “tested on CIFAR-100 test dataset and SVHN test dataset” means that we used the CIFAR-100 test dataset as an in-distribution dataset and the SVHN test dataset as an OOD dataset for the entire test dataset. We will clarify this in the revised version of our paper.
> Although the LLR performed better in Fig 3b than Fig 3a, in both cases, the LLR showed inferior performance. These results are consistent with the previous finding that LLR can assign mixed OOD scores for in-distribution and OOD samples and yield inferior performance when used with VAEs [2]. Similarly, we observed that LLR scores were not well distinguishable for OOD detection in both Fig 3a and Fig 3b. In the case of Fig 3b, LLR showed the results like random guessing (dotted blue line) although the IC performed even worse. In fact, the LLR is based on the assumption that the model trained using pure input data represents both background and semantic information, whereas the background model trained using noise-disturbed input data represents only background information. However, the encoder-decoder structure of the VAE forces it to compress the semantic information, so the model may lose some semantic information. Consequently, when used with the VAE, the LLR suffers from distinguishing between OOD and in-distribution samples.
> ***
> [1] Ren, Jie, et al. "Likelihood Ratios for Out-of-Distribution Detection." Advances in Neural Information Processing Systems 32 (2019): 14707-14718.
> [2] Xiao, Zhisheng, Qing Yan, and Yali Amit. "Likelihood Regret: An Out-of-Distribution Detection Score For Variational Auto-encoder." Advances in Neural Information Processing Systems 33 (2020).
> [3] Maddox, Wesley J., Gregory W. Benton, and Andrew Gordon Wilson. "Rethinking Parameter Counting in Deep Models: Effective Dimensionality Revisited." (2020).
> [4] Jastrzebski, Stanislaw, et al. "On the Relation Between the Sharpest Directions of DNN Loss and the SGD Step Length." ICLR. 2019.

---

> > ### Comment · Reviewer_ossS · 2021-08-29
> > **Read the response**
> >
> > I thank the authors for the clarification. This is to acknowledge that I read the authors' response. The response has sufficiently addressed my concerns, except for question 2. What I meant in the original question is that the notion of "Locally most powerful Bayesian test for OOD detction" (LMPBT) in Def 3.1 is something the authors proposed (as I understand it). It is thus not surprising that the proposed method would satisfy this. The proposed method is constructed to satisfy this to begin with. So rather than presenting only that the proposed method satisfies it, it is more interesting to see this notion explained in the context of other methods. This is why I asked
> >
> > > Is it (the proposed method) the only test that satisfies this definition?
> >
> > to which, I think, the authors have not addressed. By "implications of Proposition 1", I meant to inquire about the characteristics that the proposed method has that are not present in other methods, as a result of being an LMPBT.

---

> > > ### Author Response · Authors · 2021-08-30
> > > **Clarification of our answer to question 2**
> > >
> > > Thank you for your additional efforts and time to carefully read our response. We also very appreciate your clarification on question 2 and giving us the opportunity to clarify our answer to that question.
> > >
> > > First, we would like to clarify that although the notion of "Locally most powerful Bayesian test (LMPBT)” in Def 3.1 is formally defined by us in our paper, the concept of LMBPT is not that we creatively developed; it is a simple extension of the existing "most powerful Bayesian test" [1] to a "local" version, parallel to the relationship between the "most powerful test" and "locally most powerful test (LMPT)" [2]. That is, LMPBT shares the same rational of LMPT for maximizing the power for an alternative in the neighborhood of the null hypothesis, but in a Bayesian test framework.
> > >
> > > Second, we would like to clarify that the proposed method (the specification of $\theta_1$) is not constructed to satisfy LMPBT to begin with. The proposed method is constructed to satisfy Equation (7): we specify $\theta_1$ in the alternative hypothesis as the MLE of the expanded training set with the test sample. That is, we do not use any concept of LMPBT at the time when we construct the proposed method. Surprisingly, it turns out that the proposed method satisfies the condition of Def 3.1 and is the LMPBT as proven in Proposition 1.
> > >
> > > Third, the LMPBT property (i.e., maximizing the probability of correct detection of OOD samples) is the "unique" property of our method; other methods (such as LLR, LR, and IC) do not satisfy the condition of Def 3.1 and thus they do not enjoy such a property. Therefore, only our method guarantees to maximize the probability of OOD detection, whereas other methods do not.
> > >
> > > We hope that our response resolves your concern. If there are any further concerns or something unclear in our response, please let us know. We would be pleased to clarify those points and further strengthen our paper. We sincerely appreciate your time and efforts in reviewing our paper and your insightful comments that helped us improve our paper.
> > >
> > > ---
> > > [1] V. E. Johnson. Uniformly most powerful Bayesian tests. Annals of statistics, 41(4):1716, 2013
> > > [2] M. Omelka. The behavior of locally most powerful tests. Kybernetika, 41(6):699–712, 2005

---

> > > > ### Comment · Reviewer_ossS · 2021-08-31
> > > > **Raised the evaluation score**
> > > >
> > > > Thank you.
> > > >
> > > > > Third, the LMPBT property (i.e., maximizing the probability of correct detection of OOD samples) is the "unique" property of our method; other methods (such as LLR, LR, and IC) do not satisfy the condition of Def 3.1 and thus they do not enjoy such a property. Therefore, only our method guarantees to maximize the probability of OOD detection, whereas other methods do not.
> > > >
> > > > "only our method guarantees to maximize the probability of OOD detection, whereas other methods do not": Is this known for sure or a conjecture at this point? Regardless it would be best to add what the authors explained here in the paper. I am referring to the part that explains LMPBT in the context of other methods.
> > > >
> > > > I raised the score.

---

> > > > > ### Author Response · Authors · 2021-09-01
> > > > > **Thanks for your response!**
> > > > >
> > > > > According to Proposition 1, only our method is LMPBT and thus maximizes the probability of OOD detection, whereas other methods (LLR, LR, and IC) do not. This is for sure as proven in Proposition 1. We will clarify this in the final version of our paper. Thank you for your comments that helped us clarify the implication of Proposition 1!

---

### Official Review · Reviewer_msUZ · 2021-07-20

**Rating:** 6
**Confidence:** 4

**Summary:**

This paper proposes using a Bayesian hypothesis test for out-of-distribution detection with deep generative models.  The null hypothesis is that the newly observed data is in-sample and the alternative is that the new data is out-of-sample.  The test statistic is a likelihood ratio between the max likelihood model and the model fit to the given test input.  As this statistic is impractical to compute, the paper uses a perturbation of the MLE to find the parameters for the single sample test point.  The paper shows (Prop 1) that this test is the locally most powerful Bayesian test if the training set is sufficiently large.  Yet as the test requires computation of the inverse Hessian matrix, a low-rank approximation is necessary in practice.  Experiments are reported on benchmark image data sets (FashionMNIST, CIFAR, SVHN, CelebA, etc).  The proposed method performs well in terms of AUROC when compared to likelihood ratios and input complexity.  An ablation study of the rank of the Hessian approximation is also performs and shows the method is relatively insensitive to the rank.

**Limitations And Societal Impact:**

Yes, as detecting distribution shift can improve the safety of AI systems, this method should have positive societal impact.  There are some limitations not mentioned in the paper (e.g. the issue of identifiability), which I mention above.

**Main Review:**

### Positives

**(Mostly) well-motivated approach to detecting distribution shift**:  This paper adapts the well-founded techniques of Bayesian hypothesis testing to deep generative models.  As detecting distribution shift with DGMs, ideally, requires fast, per-point evaluation, this method seems to provide just that.  Although, see my question / critique regarding identifiability below.

**Tuning computational overhead**:  As Figure 2 demonstrates, the method seems to work well—or even a bit better—when using a low-rank approximation to the Hessian matrix.  This allows the user to tune the method's computational overhead to fit the application.  This is important as detecting distribution shift often must be done in real time.

### Critiques / Questions

**Local approximation and its validity for overparameterized models**:  While the motivating Bayesian hypothesis testing framework is undoubtedly motivated for well-identified models, it seems like identifiability could be an issue for deep generative models.  In neural networks, we know that there are often directions in parameter space around local minima that can leave the resulting model unchanged [[Maddox et al.; 2020]](https://arxiv.org/abs/2003.02139).  Thus, are parameterization issues such as this a worry when applying this test to deep generative models?  How do we know the local perturbation method won't detect these directions, thinking there is indeed distribution shift, while the model is actually invariant?

**Why is the AUROC best at around 30 eigenvalues?:**  I was surprised to see that the test's performance doesn't improve monotonically with the rank of the approximation.  Do you have any idea why this is?  You mention that the Hessian is sparse, explaining the good performance at low rank, but I don't think this would explain why performance degrades a bit as rank increases.  Could this possibly be due to the "spurious" directions in parameter space I mention above?

**Remainder term in Proposition 1**:  Prop 1, how is the remainder term resolved?  Equation 12 is stated as an approximation, and I see how the 1/n term implies a good approximation when the data set is large.  But how do we know that the remainder term also can go to zero?  It seems like the paper is concerned about the convergence of the first and second order terms without discussing when they dominate the remainder?

**Discussion of score matching and [Sharma et al. (UAI 2021)](https://arxiv.org/abs/2102.12567)**: The paper is light in its discussion of previous work on OOD detection for deep generative models, and there is no Related Work section.  I think previous work on score matching and the Fisher information approach of Sharma et al. [UAI 2021] should be discussed as I see these as related methods, using perturbations of the log density function to test for OOD-ness.

**Time Spent Reviewing:**

4

---

> ### Author Response · Authors · 2021-08-10
> **Responses to your valuable comments**
>
> Thank you for your careful review of our paper and for the insightful and constructive comments. Please find our detailed answers to your comments below.
>
> ***<Responses to "Critiques/Questions">***
>
> **1. (Local approximation and its validity for overparameterized models)**
> As you pointed out, there can be an identifiability issue for deep generative models. For overparameterized models, as you mentioned, there are often degenerate directions corresponding to the smallest eigenvalues of the Hessian, where perturbations in parameters lead to almost no difference in the results [1]. However, in our study, we can overcome this issue by adopting a low-rank approximation of the Hessian. By using a low-rank approximation with largest eigenvalues, we can prevent the parameters from being updated in the degenerate directions. That is, the use of a low-rank approximation of the Hessian allows us not only to significantly reduce the computational cost, but also to overcome the identifiability issue. In the submitted version of the paper, we focused only on the aspect of computational advantage of the low-rank approximation of the Hessian. We will revise the paper to explain another important advantage of the low-rank approximation in overcoming the identifiability issue. We believe this will strengthen the justification of the use of the low-rank approximation and help us to understand the superior performance with fewer eigenvalues.
>
> **2. (Why is the AUROC best at around 30 eigenvalues?)**
> As you mentioned, the behavior that the performance does not increase monotonically with the number of eigenvalues may be explained by an identifiability issue of overparameterized models. Because changes in the output values in the degenerate directions can be considered like random noise, the test accuracy of a compressed model can sometimes be better than that of the full model. According to [2], updating the parameters in the sharpest directions, which are corresponding to the top k eigenvectors, sometimes improves the generalization ability. Specifically, the authors analyzed the dynamics of stochastic gradient descent (SGD) in the subspaces of the sharpest directions of the loss surface (i.e., directions of top k eigenvectors of the Hessian), called Nudged-SGD (NSGD), and their influences on the training speed and the generalization ability of the final model.  The authors showed that the NSGD can optimize significantly faster and find good generalizing sharp minima, i.e., minima exhibiting better generalization performance in the loss surface, which are sharper compared to the minima found by baseline SGD. However, as discussed in [2], the structure of the Hessian is highly dependent on the dataset, and thus the impact of parameter updates in the sharpest directions on the final generalization can be different depending on the dataset.
>
> **3. (Remainder term in Proposition 1)**
> In Equation (12) for Proposition 1, the remainder term of the second-order Taylor approximation, $R(\theta_1)$, has an upper bound $\frac{M}{6}  ||\theta_1-\hat{\theta}||^3$ (Lagrange error bound), where $M$ is an upper bound of the third derivative of the log-likelihood for $||\theta_1-\hat{\theta}||<\delta$. With sufficiently small $\delta$, the first- and second-order terms will dominate the remainder term because higher order terms vanish much more rapidly, and the remainder term will be negligible. We will clarify this in the final version.
>
> **4. (Discussion of score matching and Sharma et al. (UAI 2021))**
> Thanks for pointing out the missing references. We will add multiscale score matching [3] and the Fisher information approach of Sharma et al. [4] as related work on OOD detection for deep neural networks in the final version.  Mahmood et al. [3] learned the space of in-distribution data based on norms of the score estimates of training data at multiple noise scales for distinguishing in-distribution and OOD samples. Sharma et al. [4] computed a low-rank approximation of the Fisher information matrix to characterize most influential directions on the predictions over the training data, and then estimated uncertainty of a test sample by measuring the impact of perturbations orthogonal to these directions on the prediction for OOD detection.
>
> ***<Responses to "Limitations And Societal Impact">***
> In the final version, we will discuss that the identifiability issue can be a limitation of general deep generative models and that we overcome this issue by using a low-rank approximation of the Hessian, as we answered to your first and second questions.
> ***
> [1] Maddox, Wesley J., Gregory W. Benton, and Andrew Gordon Wilson. "Rethinking Parameter Counting in Deep Models: Effective Dimensionality Revisited." (2020).
> [2]Jastrzebski, Stanislaw, et al. "On the Relation Between the Sharpest Directions of DNN Loss and the SGD Step Length." ICLR. (2019).
> [3] Mahmood, Ahsan, Junier Oliva, and Martin Styner. "Multiscale Score Matching for Out-of-Distribution Detection." arXiv preprint arXiv:2010.13132 (2020).
> [4] Sharma, Apoorva, Navid Azizan, and Marco Pavone. "Sketching Curvature for Efficient Out-of-Distribution Detection for Deep Neural Networks." (2021).

---

### Decision · Program_Chairs · 2021-09-27

**Decision:**

Accept (Poster)

**Comment:**

The paper proposes a new method for OOD detection using deep generative models based on Bayesian hypothesis testing, that they refer to as the locally most powerful Bayesian test (LMPBT). Overall, the reviewers found the paper well-motivated and the experiments support the key claims.  During the discussion phase, reviewers ossS and 6JoX increased their score and recommended acceptance. Reviewers msUZ and Huvy leaned towards acceptance but raised some concerns in the initial review; after reading the author rebuttal, I think that the authors satisfactorily address most of these concerns.

I recommend acceptance and encourage the authors to incorporate the reviewer feedback in the final version.

Additional comment:
While the paper shows that they outperform some existing methods such as IC, LR, LLR, I believe there are stronger published results (e.g.
 DoSE https://arxiv.org/abs/2006.09273) that should probably be mentioned.